# Peasant Seeds in France: Fostering a More Resilient Agriculture

**Camille Gevers [1], Helena F.M.W. van Rijswick [2,*] and Julia Swart [1,*]**

[1]   Utrecht School of Economics, Utrecht University, 3508 TC Utrecht, The Netherlands;
      c.c.r.gevers@students.uu.nl

[2]   Utrecht School of Law, Utrecht University, 3584 BH Utrecht, The Netherlands

[*]   Correspondence: H.vanRijswick@uu.nl (H.F.M.W.v.R.); J.Swart@uu.nl (J.S.);
      Tel.: +31-30-253-9352 (H.F.M.W.v.R.); +31-30-253-9413 (J.S.)

**Abstract:** The profitability of the French agricultural sector has fallen over the last two decades, leading to the suggestion of a "rupture in technical progress". Additionally, the intellectual property regime in force has contributed to the erosion of the cultivated biodiversity, limiting plant resiliency to climate change and other hazards. In the face of these challenges, agroecological farming practices are a viable alternative. This paper investigates the positive and negative aspects associated with the development of alternative seed procurement networks in France. The findings indicate that peasant seed networks can effectively contribute to overcoming many of the structural blockages with which French agriculture is confronted, but that yield concerns; higher information and supervisory costs, as well as the unfavourable legislative context, constitute key challenges to their development. However, these could be partially or totally eliminated if adequate policies are implemented. In this regard, the recommendations are to: (i) strengthen the dialogue with farmers in the shaping of policies related to the use of plant genetic resources; (ii) abrogate the "obligatory voluntary contribution" on farm-saved seeds; (iii) diversify the collection of Centres for Biological Resources, increase their number, and democratize their access; (iv) harmonize the French and European regime on intellectual property; and (v) encourage participatory research.

**Keywords:** peasant seeds; resilient agriculture; sustainability; agroecology; law; biodiversity

## 1. Introduction

Modern agriculture is at a tipping point. In France, for over two decades production levels have stagnated, while productivity gains have fallen despite successive reforms [1]. In parallel, the increasing precariousness of cultivation [2], and the accelerating degradation of biodiversity [3] suggest that today's agricultural system ought to be rethought. Between the order for quicker and cheaper production in the post-war period and the aspiration to produce better and sustainably in the future, French agriculture is at a crossroads.

A new agricultural system based on a social contract that reconciles profitability, ecology, and humanism must be envisaged. In this regard, the practice of agroecology (According to the Food and Agriculture Organization of the United Nations (FAO), agroecology is "a scientific discipline, a set of practices and a social movement. As a science, it studies how different components of the agroecosystem interact. As a set of practices, it seeks sustainable farming systems that optimize and stabilize yields. As a social movement, it pursues multifunctional roles for agriculture, promotes social justice, nurtures identity and culture, and strengthens the economic viability of rural areas". Source: http://www.fao.org/family-farming/themes/agroecology/en/) has been increasingly credited by the scientific community as a solution to overcome structural blockages [2,4,5]. Yet, transitioning

to agroecology on a national level implies reconfiguring legal and economic assumptions through which the current system has been legitimized. In particular, agroecology begs the question of seed sovereignty, which refers to a set of core principles shared by farmer movements across the globe, with the common objective of redefining plants' genetic heritage and biodiversity as public goods. Although divergences exist on the meaning of this semantic appellation [6], proponents of seed sovereignty share four similar claims: (1) the right to save and replant seeds; (2) the right to share seeds; (3) the right to use seeds to breed new varieties; and (4) the right to participate in shaping policies for seeds [7]. By extension, agroecology begs the question of the rehabilitation of peasant seeds.

In French, *semences paysannes* (peasant seeds) designate, according to the definition of the *Réseau Semences Paysannes* ("French Farm Seed Network", which is the main association promoting the use of peasant seeds in France, see http://www.semencespaysannes.org.), seeds that are: (1) derived from a dynamic population or set of population; (2) reproducible by the cultivator, selected and multiplied with non-transgressive methods of the plant cell, and at the reach of the final cultivator, in peasant, organic, or biodynamic agriculture (all three concepts closely relate to agroecology. Peasant agriculture is not a type of agriculture, per se, but a set of values. Its principles are synthetized in the FADEAR's (Network for Peasant Agriculture) Charter of Peasant Agriculture, which stresses, among other things, the need for transparency in the food production chain, the maximization of holdings' autonomy, long-term planning, and North–South solidarity. More information can be found at: http://www.agriculturepaysanne.org/la-charte-de-l-agriculture-paysanne. Organic agriculture refers to a concrete farming practice that bans the use of artificial inputs (fertilizers, pesticides, fungicides, etc.). Biodynamics is an esoteric approach to agriculture, which considers holdings as autonomous living organisms, and rests on lunar and planetary cycles); (3) renewed by successive selection or multiplication in free pollination, without self-fertilization over multiple generations; and (4) freely exchangeable in the respect of conditions defined by the cooperatives that make them thrive [8]. Importantly, peasant seeds must be distinguished from farm-saved seeds in that the latter ones, though selected in situ, are derived from commercial varieties, thus certified by public authorities, and most often, protected by intellectual property rights. Additionally, while peasant seeds can be used in organic farming, organic seeds need cannot be peasant seeds. Commercial varieties of organic seeds are currently subject to roughly similar regulations to the conventional seed market [9].

The economic consequences of gratuitous seed exchanges, legalized in France since 2016 (Loi pour la reconquête de la biodiversité, de la nature et des paysages, article 12) (Article 12 of the "Law for the recovery of biodiversity, nature, and landscapes" (translation from the authors) modified the Rural and Sea Fishing Code to enlarge the authorization of reciprocal seed exchanges to all cultivators (2018, article L315-5). This prerogative was, until then, only reserved to specific farmer associations, called "Groups of Specific Economic and Environmental Interests" (translation from the authors)), have been virtually ignored in the economic literature. Admittedly, transitioning to more sustainable agricultural practices has raised the interest of many researchers in economics [10–12]. Yet, little attention has been paid to the economics of seed procurement, management, and conservation. Moreover, the few pioneering research studies that have recently been conducted in this domain concentrate on alternative commercial approaches.

In the United States [7] there have been proposals to design a model for seed procurement directly inspired from the open source software movement: the Open Source Seed Initiative (OSSI). However, the OSSI's vision of "biological open source" translates into the promotion of an exclusory "open source license", for which adherents must pay a fee [7] (p. 1226).

While challenging the soundness of the privatization of vegetative material, this nascent field of research does not question the commodification of seeds itself. In fact, seed laws have encountered more nuanced criticism in environmental sciences, law, anthropology, ethics, and philosophy [4,13–15]. By investigating the promises and challenges of scaling up the use of peasant seeds in France in the context of the agroecological transition, this article aims to contribute to the economic literature in two ways. First, the paper explores non-monetized seed exchanges, a perspective which echoes their status

of "common heritage of humankind", such as defined by the Food and Agriculture Organization [16] (p. 165). Second, the paper concentrates the analysis at the national level, in order to shed light on the political-economic dynamics at stake in a particular territory. The choice of France is explained by its strategic position on the global seed market. France is the largest European producer of seeds, and the top global exporter [17]. Additionally, France has a rich history of defending peasant values.

Our paper contributes to other studies that analyze the development of peasant seeds throughout the world. One previous study [18] investigated peasant movements in Europe and the possibilities of the movement spreading out towards Southern countries. Another study [19] argued that this recent movement of peasant seeds in Europe is an attempt for recognition and cognitive justice. The authors give support to this argument by conducting a qualitative investigation of an initiative in France that has set up a collective breeding program. Another study [20] analyzed participatory plant breeding in France and showed that it is suited to organic farming in France.

The paper begins by drawing the current portrait of French agriculture, and by highlighting its structural challenges. The empirical analysis is based on time series data from the Farm Accountancy Data Network (FADN), and the National Institute of Statistics and Economic Studies (Insee), ranging from 2004 to 2016. The focus of the quantitative analysis is on field crops, which occupy the vast majority of the total cultivated area (92% in 2016, according to the data of the FADN). Thereafter, the paper exposes the principles and limits of peasant seeds, as well as their status in the French and European legislation. The paper concludes with policy recommendations and a brief conclusion.

## 2. Methods

In order to investigate the positive and negative aspects of peasant seeds in France, the paper takes an interdisciplinary approach by combining a quantitative analysis of recent time-series data (available for the years 2004–2016) related to agricultural productivity, prices, costs, revenues, and indebtedness with a description of the legal perspective. For this purpose, the paper describes the current French Agricultural landscape, combining data analysis with findings from the literature. The main data source is "European Commission—EU FADN".

Thus, the paper links the data analysis to an extensive interdisciplinary literature review on peasant seeds and focuses on legal aspects in France and Europe that define how seed production and use are regulated in France. The paper extends the analysis to the present time, where France appears as a pioneering country with a government calling for projects to support initiatives promoting sustainability of collections of plant genetic resources for agriculture. By looking at these recent advances and taking this interdisciplinary approach the paper sets policy recommendations for France.

## 3. Results

### 3.1. The French Agricultural Landscape

Since the post-war period, considerable productivity gains have been realized in France and Europe in a context of profound mutation of the agricultural world. Over the last forty years, the substitution of manual labor for machinery and chemical inputs has enabled low food prices to be secured [21]. Furthermore, the emergence of bigger holdings has allowed the realization of significant economies of scales, and led to the specialization of production at regional levels [2].

The advent of industrial agriculture was the result of a desire from European leaders to achieve food security as rapidly as possible, at an epoch where the fear of political instability, geopolitical imbalances, and material scarcity were looming. France was, for example, importing half of its foodstuff in the 1950s [2]. In 1958, the Common Agricultural Policy (CAP) was founded, and became an emblematic measure of inter-state cooperation, foodstuff price-controls, and agricultural subventions. Its objectives, enshrined in Treaty of the Functioning of the European Union (TFEU) [22], bore witness to the productivist and security ideology of the time, proposing that: agricultural productivity shall be increased by means of "technical progress" and "the optimum utilization of the factors of production";

markets shall be "stabilized"; supplies made "available" and "at reasonable prices"; while a "fair standard of living for the agricultural community" shall be guaranteed (2012, article 39). The policy bore fruits. By the 1980s, France had reached food self-sufficiency [2].

Thus, in France, the 20th century was characterized by industrialization in the agricultural sector with a focus on increasing productivity. As a result, crop production and breeding seeds became two separate activities. Farm seeds are still available, nonetheless the tendency is for a decrease in its share, unless awareness of the problems accompanied by the increase in seed industries and seed standardization grows and policies to revert this trend are implemented. In the Discussion section we present evidence of the emergence of a political movement for change. The EU Council Directive 98/95/EC (Retrieved from https://eur-lex.europa.eu/legal-content/EN/TXT/PDF/?uri=CELEX: 31998L0095&from=en) was a first step to recognize the threat to genetic loss and the connection to the market of seeds. Also, at the European Union level, in more recent years new directives have encompassed seed conservation (e.g., Directive 2008/62/EC and Directive 2009/145/EC) (Retrieved from https://eur-lex.europa.eu/legal-content/EN/ALL/?uri=CELEX:32008L0062 and https://eur-lex.europa. eu/legal-content/EN/ALL/?uri=CELEX:32009L0145, respectively)). France's position with respect to these Directives was in favor of maintaining both a national catalogue and a common catalogue of seeds, where the first is necessary to take into account national considerations with respect to climate and plant disease risk. The exchange of seeds, on the other hand, was legalized in France in 2016. Nonetheless, the French and European legal framework remains unfavorable to the development of seed commons (see Section 3.3). The following sub-sections analyze economic, social, and ecological aspects associated with peasant seeds in France.

### 3.1.1. Productivity

As the new the millennium approached the first signs that the system was running out of steam appeared. In France, the volume of agricultural production, as well as the productivity of intermediary consumption, started to stagnate around the middle of the 1990s [1]. During the preceding decades, the development of biogenetics had encouraged farmers to widely rely on artificial inputs (high yield cultivars (plant varieties with a narrow genetic base that are derived from selective breeding methods, such as pure line selection (i.e., the self-pollination of plants with the aim of obtaining a homozygous progeny)), fertilizers, pesticides, fungicides, etc.), of which productivity was rising at an annual rate of 0.5% [1]. However, the beneficial effects of these changes of practice quickly lost momentum, and led to the suggestion of a "rupture in technical progress" [1] (p. 3), i.e., the possibility that genetics and nitrogen chemistry could not address the challenges of the new millennium (climate change, degradation of the biosphere, increasing demographic pressure). In effect, since the middle of the 1990s, inter-annual output variations have become central [2]. Between 2004 and 2016, the yield of maize and wheat (France is the largest producer of wheat and maize within the European Union (see https://ec.europa.eu/eurostat/statistics-explained/index.php/Main_annual_crop_statistics, retrieved on September 07, 2018)), for example, was significantly lowered (−12% and −32%), at an average annual rate of −1% and −3%, respectively (see Figure 1).

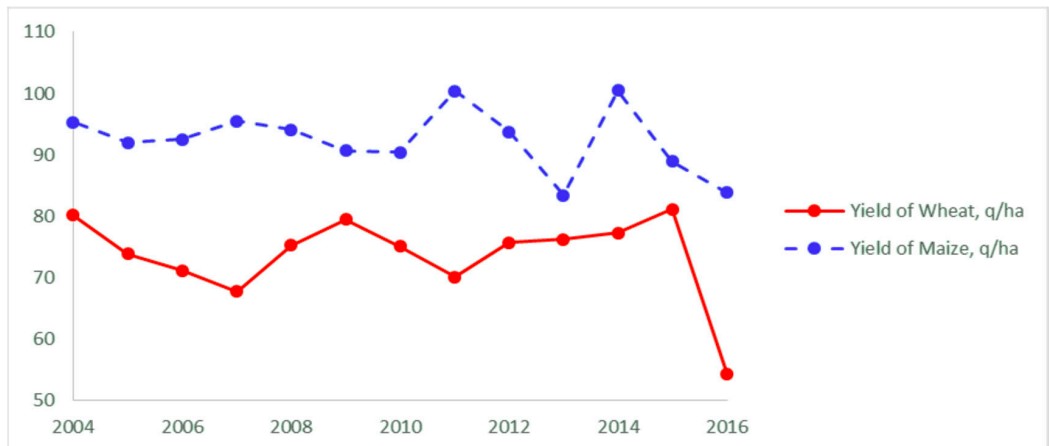

**Figure 1.** Wheat and maize productivity in France, 2004–2016. Notes: Own calculations, based on data from the Farm Accountancy Data Network (FADN). Productivity is measured by output per hectare.

### 3.1.2. Price Volatility

In parallel, the heightening of foreign competition induced by the liberalization of global trade and the 1992 reform of the Common Agricultural Policy (CAP) (for the sake of competitiveness, European leaders decided at the time to align the price of European foodstuffs to world market prices) dragged food prices down in the beginning of the 1990s [23], exacerbating farmer's pressure to manage production costs more tightly in order to keep their holding afloat [24]. As an example, between 1991 and 2004, the price paid to producers for cereals and oleaginous plants fell by 4% to 6% every year in constant euros [24], to the point that farmers periodically operated at a loss in the production of most field crops. In more recent years, food prices have increased again, albeit with high volatility (Figure 2). Price volatility, in turn, implies first that producers have to operate under high uncertainty, an inhibitor to longer-term investments. Second, price volatility represents that producers face uncertainty with respect to their own income.

In addition, the financialization of raw agricultural products has materialized into important price fluctuations over the last ten years. Financialization, by abstracting "food from its physical form into highly complex agricultural derivatives", and increasing the number of actors in commodity chains, contributes to price volatility [25] (p. 797). Since the 2008 crisis, the phenomenon has gained such importance that a recent ministerial report identified this volatility as a threat to the stability of the farmers' profession, and food security in general [26].

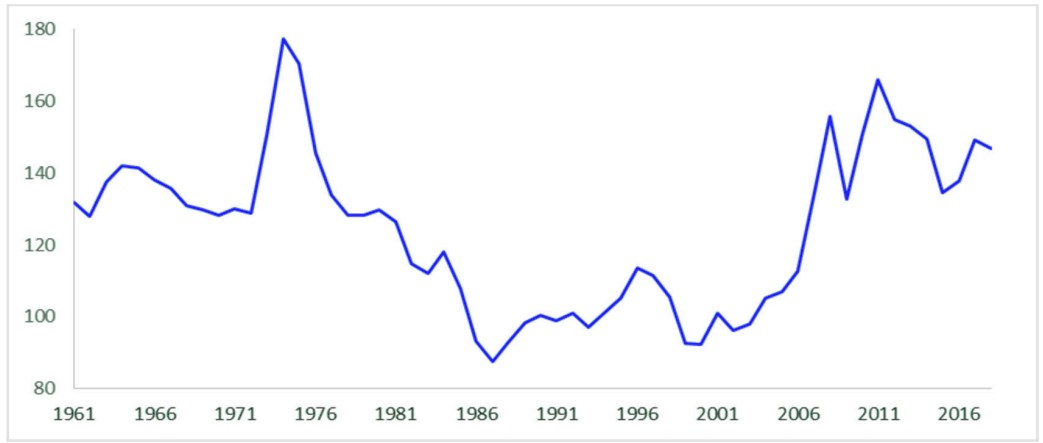

**Figure 2.** Food price index (real prices, 2002–2004 = 100), 1961–2018. Notes: Based on Food and Agriculture Organization of the United Nations (FAO) data, retrieved on September 25, 2018.

### 3.1.3. Higher Crop Costs

Our analysis shows that in addition to this sustained market pressure, the field crop sector faces rising bills in intermediary consumption (+24% between 2004 and 2016, in constant euros). More specifically, the seed cost per hectare has jumped by 12% (2004–2016, in constant euros), which is tantamount to an average annual increase of 1% (see Figure 3). This finding is also consistent with data from the Insee, who publishes an index of seed cost, the Agricultural means of production purchasing price index (IPAMPA), illustrating the same trend (see Appendix A, Figure A1). Assuming that the density of seedlings has remained constant over the period, this increase can only be justified by the growing market concentration of the seed industry. Furthermore, the structure of the agricultural sector is such that it creates strong lock-in effects. As recently noted in a report by the European Parliament, "considering that seeds are becoming increasingly bred for dependence on other inputs" (Bové, 2011; cited in [24] (p. 5)), the combined increase in seeds cost, synthetic fertilizers, and plant protection products places farmers in a particularly vulnerable position. Over the period of interest, fertilizer costs have soared by 42%, while crop protection products have risen to a lesser extent by 13% (see Appendix A, Figure A2). It is also worth stressing that many of the biggest breeding companies, such as Monsanto, Bayer, and Syngenta, are heavily engaged in the market of phytosanitary products and fertilizers.

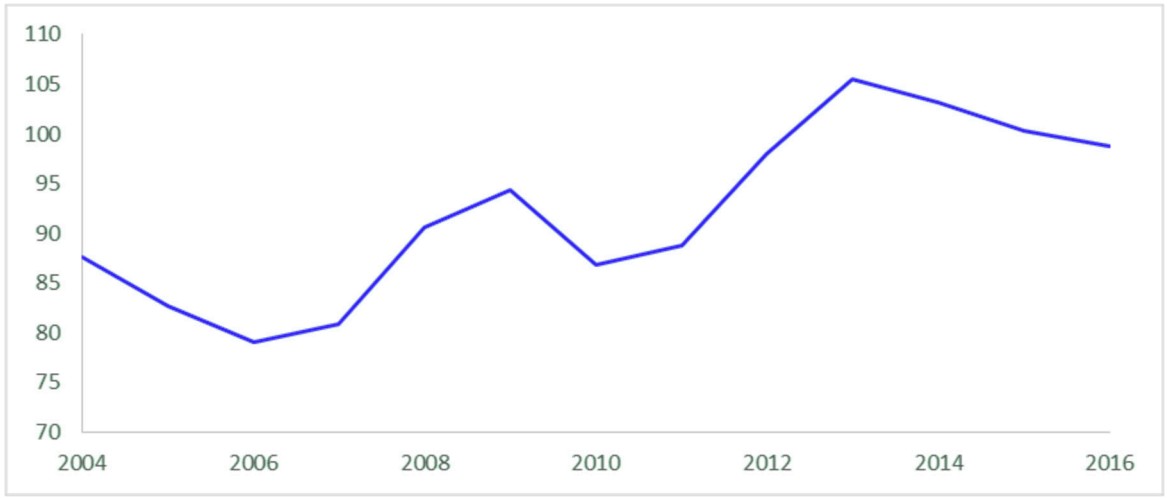

**Figure 3.** Seeds and plants costs per hectare in France, 2004–2016. Notes: Own calculations, based on data from the FADN. Based on constant euros data (2004 = 100).

France counts 565 seed businesses within its territory, of which only 72 of them are breeding companies [24] (it is, nonetheless, not possible to evaluate their independence, as some of them belong to the same group (e.g., Clause, Erodur, Limagrain Europe, and Vilmorin SA are all part of Invivo)). Out of these, three alone (Limagrain, Dupont, Syngenta) control 47% of the market [24] (estimates from 2011.). Furthermore, the stronghold of these seed giants is most predominant in the economically most-important species, such as maize, cereals, and potatoes [24].

### 3.1.4. Revenues and Indebtedness

The conjunction of the three phenomena discussed above (faltering productivity, volatility in agricultural commodities prices, and rise in input costs) has weighed heavily on the standard of living of cultivators, as well as the profitability of the sector. Between 2004 and 2016, revenues in field crops plummeted by 82% (barely reaching 5215 Euros annually at the end of the period, see Figure 4). Moreover, farmers have been operating with losses for most years (see Figure 5), de facto depending on subsidies as their primary source of income.

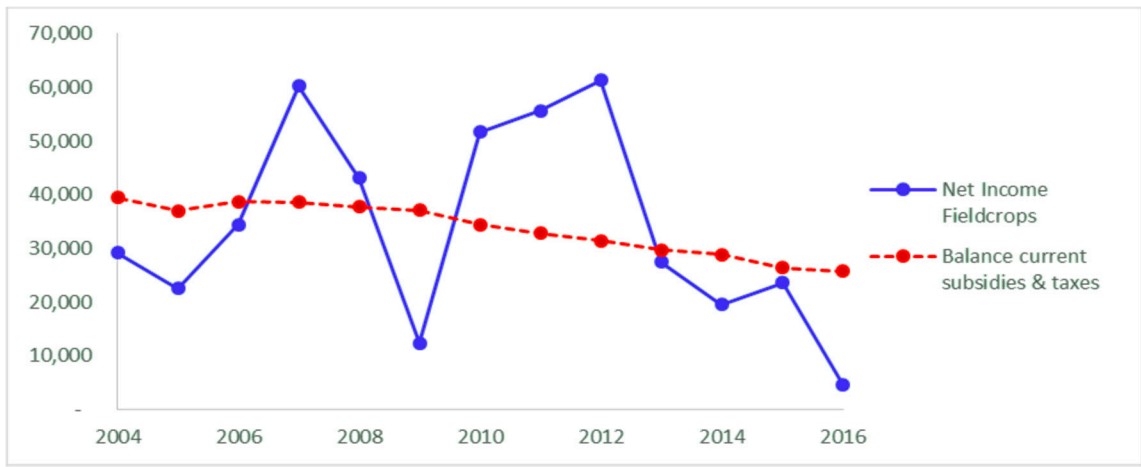

**Figure 4.** Field crops net income and balance subsidies and taxes in France, 2004–2016. Notes: Own calculations, based on data from the FADN. Based on constant euros data (2004 = 100).

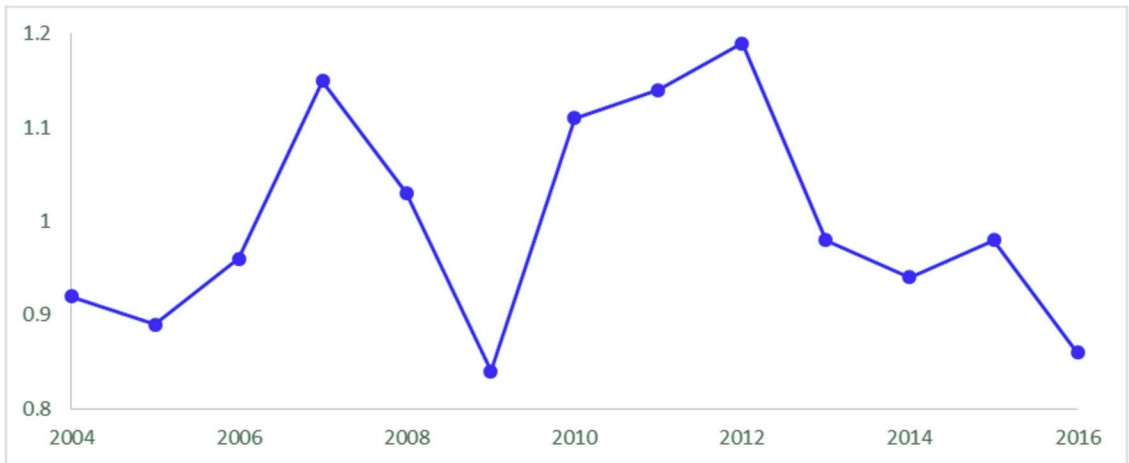

**Figure 5.** Field crops output relative to inputs (in euros) in France, 2004–2016. Notes: Own calculations, based on data from the FADN.

At the same time, the average debt ratio has increased by three percentage points in the period 2004–2016 (from 42% to 45%; see Figure 6), driven mainly by an increasing trend in medium and long-term liabilities (+40% in the same period, see Figure 7). This latter observation reveals a profound contradiction in the agricultural world—cultivators are encouraged to make growing investments in land and infrastructure, endlessly forging ahead, whereas the profitability of their holding itself is faltering. Land acquisition has increased by 38% over the period 2004–2016 in constant euros. The phenomenon is also true for buildings, though the increase is more moderate (see Appendix A, Figure A3).

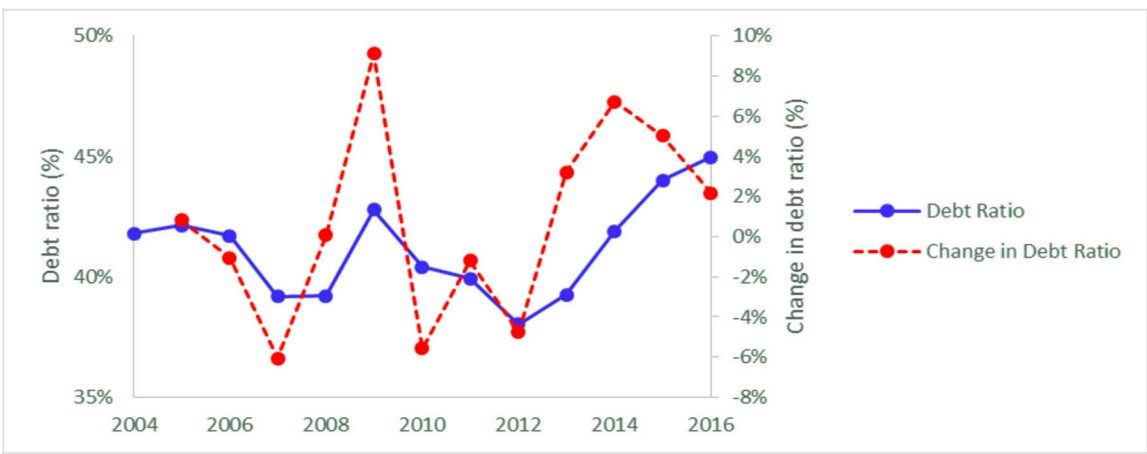

**Figure 6.** Field crop debt in France, 2004–2016. Notes: Own calculations, based on data from the FADN.

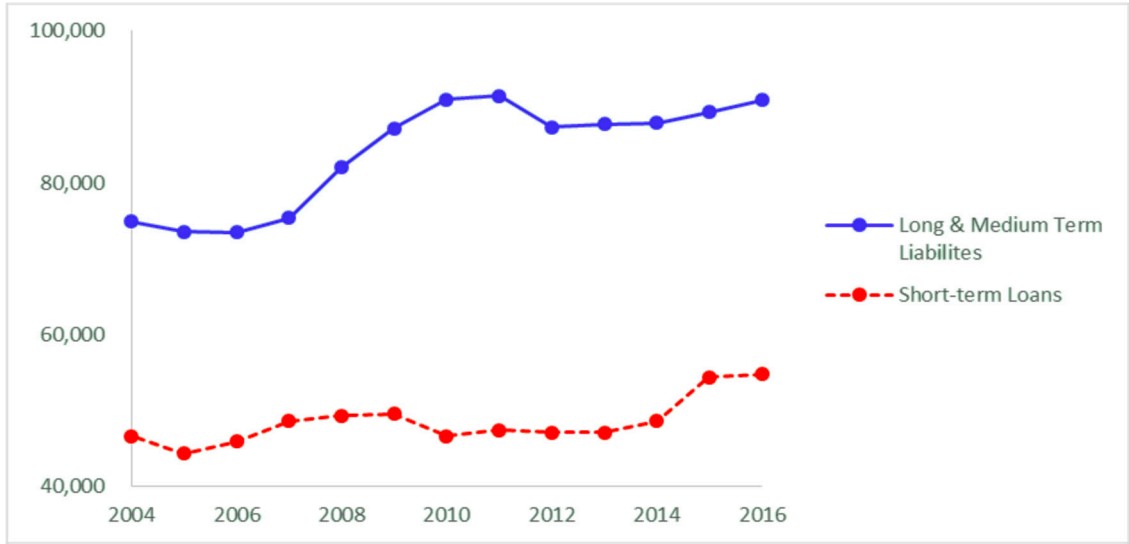

**Figure 7.** Field crop drivers of indebtedness in France, 2004–2016. Notes: Own calculations, based on data from the FADN.

The financial strain to which farmers are exposed has disquieting implications. France has witnessed, for example, a wave of suicides among the agricultural population, exacerbated by the 2008 crisis, which some activists estimate in the order of six hundred deaths per year [27]. Whereas official numbers fluctuate at around two hundred deaths per year, activist groups argue that methodology concerns lead government officials to underestimate the extent of the phenomenon.

As cultivators are trapped between the order to continuously augment their production and the growing evidence that their economic status is deteriorating, the sustainability of the profession itself has become endangered. The demographic decline of heads of holding in 2017 was the most marked of the decade, whereas their mean age inexorably progressed [28]. The number of heads of holdings declined by 1.9% in 2016, whereas their mean age moved from 49 in 2016 to 49.3 years in 2017. Moreover, according to another survey, one head of holding out of five was over 60 in 2010 [28]. In addition, 30% of the medium and large-size holdings had no designated successor in 2010, a number which rose up to 50% for small holdings [29].

### 3.1.5. Erosion of the Cultivated Biodiversity

Lastly, the industrial agricultural model could eventually compromise food security, the very purpose it was designed to achieve. In 50 years, the range of field crops commercialized in the French

Official Catalogue (the French Official Catalogue gathers all the varieties authorized for marketing in the national territory. Its management falls within the competence of the Ministry of Agriculture, which is supported by the Permanent Technical Committee of the Selection of Cultivated Plants (CTPS). The Catalogue is accessible online, see https://www.geves.fr/catalogue/) increased from around 500 varieties in 1960 [30] to 3252 today. This impressive number, regularly put forward by the seed industry, renders a false sense of diversity. In fact, cultivated species have become increasingly homogenous, a consequence of decades of research focused primarily on yield improvement [30]. In fact, this trend has been observed globally. The FAO estimates that the diversity of cultivated crops shrank by 75% over the last century, further warning that a third of today's diversity could disappear by 2050 (FAO, 1997) [24]. According to the same institute, whereas 7000 species have been used throughout the history of mankind to feed populations, only 30 account at present for 90% of our nutritional intake, and only three crops (rice, wheat, corn) constitute more than half of our diet (FAO, 1997) [24]. The homogenization of germplasm also corresponds to the requirements of industrial agriculture, imposed by the standardization of output and the mechanization of agricultural practices. Moreover, it has been widely documented that the increasing market concentration of the seed industry has exacerbated this phenomenon [31–33].

Although homogenization has allowed the expression of undesirable genes to be eliminated, it has also made cultivated varieties more vulnerable [31]. In effect, genetic diversity is of primordial importance to increase plants' resiliency to climate change and other hazards [34]. Far from being long-term preoccupations, these adverse shocks have begun to be felt. The narrowing of crops' genetic base has already translated into a loss of resistance to pest attacks, hence a greater use of phytosanitary products [35]. In parallel, the abnormally low results of 2016 have been attributed to the lack of crop adaptability to more frequent droughts [17].

### 3.2. Principles and Limits of Peasant Seeds

The recent performance of French agriculture encourages us to question our measure of progress. Admittedly, the absolute priority given to yield improvement has helped France to reach food self-sufficiency. However, the structural blockages with which France is confronted shed light on the necessity to take account of the social and environmental realities of the agricultural sector. In this regard, the use of peasant seeds has been advocated by many intellectuals as a way to reconcile profitability with crop stability and farmers' integrity [36–39]. Nonetheless, peasant seed networks are subject to several limitations, which may not be overlooked. The following section is an analysis of the positive and negative aspects of these alternative systems.

#### 3.2.1. Food Sovereignty and Biodiversity Conservation

The role of peasant seeds in preserving biodiversity and ensuring crop stability is acknowledged by international law. One of the distinctive characteristics of the so-called "peasant" systems is that the process of domestication and variety improvement is led by cultivators themselves through the continuous selection of plants most-adapted to local farming constraints. Landraces (a landrace is a population of cultivated plants that has not been subject to formal crop improvement but is rather associated with the tradition and knowledge of the people who grows it. In French, landrace translates into "*variété paysanne*" (peasant variety), hence the appellation "peasant seed") are, therefore, characterized by greater heterogeneity, which makes them more resilient to adverse shocks. This latter feature has contributed to the resurgence of interest in landraces in scientific research, as their adaptive traits are increasingly being documented as a way to cope with the effects of climate change [39,40].

#### 3.2.2. Encouraging Social Link and Lowering the Bill

In addition to their greater genetic richness, peasant seeds may, through the development of seeds commons, serve as a catalyst of social links. In France, the Réseau Semences Paysannes re-groups 38 peasant seed houses (« *Maisons des Semences Paysannes* », translation from the authors) [41]. While an

exhaustive analysis of their mode of organization falls out of the scope of this paper, next follows an exposition of their fundamental principles.

Firstly, we discuss the advantages of mutualizing genetic material. Peasants pool resources to improve their means to conserve landraces, select new varieties, and develop their agricultural skills [42]. Behind the creation of every seed house lays the prospect of favoring the exchange of practices, increasing horizontal synergies, giving a sense of empowerment, and providing a place for listening and support. These motives highly contrast with the individualistic mentality of the dominant paradigm, which tends to isolate and reduce farmers to the status of self-centred agents, competing for more land and resources. In this light, seed commons may, by virtue of creating room for dialogue and mutual assistance, be a first step towards reducing farmers' psychological burden.

Seed commons may also reduce upfront costs and prevent dangerous debt cycles. According to our analysis, farmers in conventional agriculture spent over 114 Euros per hectare in seed purchase in 2016, a cost that has been increasing at an annual rate of 2.26% over the last decade (2004–2016). On average, this expense represents 11% of holdings' total intermediate consumption (2016). For a typical average holding of 117 hectares (2016), free access to seeds would represent a rough saving of 13,338 Euro per year. This saving would have a significant impact on farmers' net revenues (+225% with regard to the 2016 level, +32% for the 2004-2016 average). Moreover, this amount represents almost 8% of holdings' total liabilities (2016). The cultivation of landraces is also less input-intensive [43], which could lessen farmers' dependence on ever more expensive chemical products.

### 3.2.3. Shortages and Transaction Costs

The peasant selection, nonetheless, contains inherent limitations. In alternative systems, cultivators are more prone to generalized seed shortages, as they tend to rely on local exchange networks, which can be subject to adverse shocks [44]. On the other hand, wider networks increase the risk of fraudulent behavior, as the quality of peasant seeds is not certified by institutional controls [45] (in the French conventional system, the quality of seeds developed by plant breeders is examined by the National Authority of Plant Variety (INOV), a prerequisite for being listed in the Official Catalogue. Depending on their aims, plant breeders may also opt for a European-wide protection, by submitting their innovation to the Community Plant Variety Office (CPVO). See http://cpvo.europa.eu/en). Peasants, therefore, face a trade-off between diversification and supervisory costs, and collective organization calls for a reflection on sanitary matters. Importantly, the decentralization of quality controls need not happen at the detriment of the final consumer, as in any case, peasants internalize the risk of sanitary problems—damaged seeds impact crop yield, which ultimately affect peasants' revenues. In order to minimize this risk, members of seed commons traditionally resort to three different strategies (which are not mutually excludable): the drafting of a charter; the setting up of a system of internal control (germination test, detection of genetically modified organisms, GMO, technological analysis, etc.); and the creation of training sessions [42]. This latter measure is especially indicated as a means to reduce information costs. Indeed, internalizing breeding activities requires expertise and technical skills, which represent a major hindrance for the uninitiated [40]. In the mainstream system, by contrast, specialized scientific entities are entrusted with the development of new varieties, for which they receive intellectual property rights. Farmers are relegated to the status of "licensee", paying royalty fees in order to get the authorization to use plants' genetic material.

### 3.2.4. Yield Concerns

The comparative yield of landraces and commercial varieties is the subject of controversy. Nonetheless, it is generally acknowledged that landraces typically lag behind commercial varieties in terms of profitability [46]. Peasant seeds' lower productivity is partly explained by the fact that genetic richness augments intra-specie competition, which negatively affect output [47]. Landrace performance is, therefore, highly sensitive to the density of seedling, crop association, and other agricultural practices [48], which makes their comparison to industrial crops highly complex.

To the best of our knowledge, there exists no comprehensive study on the comparative performance of landraces and commercial varieties, whether in organic or conventional farming. However, in the context of organic farming, landraces may equal or outperform commercial varieties. In a previous study of the performance of two wheat landraces [47] (p. 1), it was observed that "although (the landraces) lagged behind the cultivar by 64% and 38% at the dense stand, the reverse was true with spaced plants, where they succeeded in out-yielding the cultivar by 58% and 73%, respectively", and it was concluded that the potential of landraces might be systematically undervalued under a dense stand. Another study conducted on Lucerne crops in Italy showed no significant yield difference between landraces and cultivars [49]. Similar results are confirmed in a Hungarian study on the assessment of landraces of tomatoes, where the authors noted that "the yield of each landrace (was) greatly influenced by conditions and technology on the farm" [50].

In response to this nascent field of research, new landrace improvement techniques are brought forward [51]. Among these, the development of agroecological practices for minimizing intra-specie competition, the identification of the adaptive traits of landraces, as well as their understanding on a molecular basis seem to be promising avenues of research. In this respect, participatory plant breeding programs, where researchers work in close collaboration with peasants, have proven to be particularly efficient [51].

### *3.3. Peasant Seeds and Intellectual Property Rights*

Although the exchange of seeds has been legalized in France since 2016, the French and European legal framework remain unfavorable to the development of seed commons in several ways. In order to fully seize the extent of these legislative challenges, one should note that the regulation of seed production and use does not pertain to a single legal source, but to a collection legislative packages (intellectual property rights, commercialization standards, phytosanitary norms, and biosecurity). Next follows a discussion of the evolution of the intellectual property regime in France and Europe, for this legal field has a direct influence on the creation of peasant seed houses.

### 3.3.1. The Origin and Shortcomings of Plant Variety Rights

Economists classically justify the presence of intellectual property rights on the ground that a temporary monopoly on an innovation allows its creator to gain a return on investment, without being confronted by competitors. In Europe, such rights were established in 1961 with the international convention of the Union on Plant Variety Certificates (UPOV) (whereas the initial signatory members were all European, the UPOV successively enlarged to count today of 75 acceding countries [52]). At that time, the signatory members aimed to create a harmonized system of intellectual property rights specific to vegetal varieties, which materialized by the creation of the Plant Variety Rights (PVR).

PVR grant breeders intellectual rights on the base of: (i) newness; (ii) distinctiveness; (iii) uniformity; and (iv) stability [53] (article 5(1)), for a duration of 20 to 25 years (article 19). Although, on one hand, this harmonization encouraged agronomic innovation, it set, on the other hand, the legal basis for the marginalization of peasant seeds, a phenomenon [15] that has metaphorically been called the "monocultures of the mind". In effect, the more diverse phenotypical characters of the peasant seeds preclude them from being evaluated, and hence commercialized, according to strict criteria. As a matter of fact, among the five thousand plant varieties commercialized in France today, the six hundred species protected by a PVR represent 99% of the cultivated biodiversity (Guey, interviewed in a previous study [54]).

PVR slightly differ from patents in that the first Convention of the UPOV 1961 restricts the enforcement of intellectual rights when acts are done for the "purpose of creating other new varieties" (breeder's exception) (article 5(3)). (This derogation is restated in UPOV 1991 as "acts done for experimental purposes" and "acts done for the purpose of breeding other varieties" [53] (article 15(1)(i) and (ii))) Moreover, UPOV 61 implicitly excludes the on-farm selection and reproduction of seeds from the scope of application of PVR (farmer's exception) (article 5(1)). (Article 5(1) states that "the

effect of the right granted to the breeder of a new plant variety or their successor in title is that their prior authorization shall be required for the production, for purposes of commercial marketing, of the reproductive or vegetative propagating material, as such, of the new variety, and for the offering for sale or marketing of such material" [55]. The clause omits to extent breeder's rights to harvested materials) At the time, the breeder's exception was intended to lower barriers of entry into the seed industry and spur research in the development of high yield cultivars. As for the farmer's exception, the techniques of the epoch could not determine, without an onerous expertise, which varieties were derived plants that had been successively reproduced by farmers [3]. Hence, it is primarily for lack of technological means that intellectual property rights were not extended to on-farm selection.

However, as years passed, scientific discoveries gradually strengthened the case of the seed industry. With the development of biotechnologies, microbiological processes—which escape the modalities of PVR (patents on genes are delivered by the European Patent Office, accessible via https://www.epo.org/index.html. Biological materials are only patentable if not obtained via essentially biological processes [56] (article 4(b), and must satisfy, besides the general criteria of patentability (newness, inventive activity, and industrial application), two other conditions: (i) traits must be isolatable (article 3); and (ii) their industrial application must be concretely exposed (article 5(3)). Moreover, the directive allows genes existing naturally to be patented—"biological material which is isolated from its natural environment or produced by means of a technical process may be the subject of an invention, even if it previously occurred in nature" (article 3 (2)))—enabled breeders to assert ownership over already protected varieties, by adding patented genes to their germplasm. In the face of this illegitimate appropriation, the reform of the UPOV in 1991 extended the protection of PVR to varieties "essentially derived" from protected varieties [53] (article 1(5)(a)(i)). In spite of this, the entanglement of patented genes and PVR on given cultivars has pushed research budgets up, reinforcing the dominant position of a few well-established businesses [43].

Additionally, the development of molecular marking in the 1980s facilitated the identification of patented genes, allowing them to be effectively traced in agricultural products, on the basis of a simple sampling [32]. As the seed industry possessed the technological tools to sanction farmers, the progressive instauration of a pejorative lexical field as regards farm-saved seeds convinced the international community to extent the scope of PVR to the agricultural production of protected varieties, and to the fruit of their harvest, effectively confirming the death of the farmer's exception [53] (article 15(1)(i) and article 16(2)). In fact, seed breeding unions commonly refer to farm-saved seeds as "black seeds", implying that the on-farm reproduction of protected varieties is a form of "free-riding" and "counterfeiting" [57]. If, on paper, the farmer's exception subsists [53] by virtue of article 15(1)(i), namely when acts are "done privately and for non-commercial purposes", this right becomes widely limited— by enlarging the scope of the breeder's right to "harvested material" or "any product made directly from the harvested material" (article 16(2)(ii) and (iii), respectively) [53], sets the legal basis for the criminalization of farm-saved seeds. A few years later, the European Union condemned farm-saved seeds as a form of patent infringement, holding cultivators liable to pay "an equitable remuneration" to the holders of varieties reproduced on-farm [58] (No. 2100/94, 1994, article 14(3)). France has been, thus far, the only Member State to transpose this clause into its national legislation, with the establishment of a "compulsory voluntary contribution" (*Contribution volontaire obligatoire*", translation from the authors) of 21 varieties [59] (article 16), ("Law Relative to Plant Variety Rights", translation from the authors), a measure which concerns 50% of the farming profession [40].

Advances in biotechnologies have been infringing on farmers' autonomy for another reason. Developments in European case law have legitimized the use of molecular marking as a means to acquire ownership over any sequence of germplasm, or the information it contains, despite the fact that this technique in no way isolates, transposes, or modifies the genes (or information) of interest. In the so-called "Tomato II and Broccoli II' cases (G 0002/12 and G 0002/13, 2015), the European Patent Office Board of appeal ruled that patents could be granted to plants, vegetative materials, or information obtained through essentially biological processes, provided that the fundamental

conditions of patentability were present. As a matter of fact, the European legislation has failed to anticipate the possibility of native gene (characteristics that wild or cultivated plants possess naturally, which are the result of biological processes, or natural phenomena (e.g., crossbreeding, selection)) patenting, leading to a risk of "biodiversity totalitarianism" [60]. In France, the risk of native genes patenting has been mitigated by the interdiction since 2016 for French seed breeders to claim property on any variety obtained through essentially biological processes, including their *constitutive elements* and *genetic information*, irrespective of the use of molecular marking [61] (article 9). Article 9 introduced a modification in the Code of Intellectual Property so as to forbid the patentability, not only of varieties derived from essentially biological processes, but also of the genetic material or information they contain: « *Ne sont pas brevetable: (3) Les procédés essentiellement biologiques pour l'obtention des végétaux et des animaux; sont considérés comme tels les procédés qui font exclusivement appel à des phénomènes naturels comme le croisement ou la sélection (3^{bis}) Les produits exclusivement obtenus par des procédés essentiellement biologiques exclusivement obtenus par des procédés essentiellement biologiques ( . . . ), y compris les éléments qui les constituent et les informations génétiques qu'ils contiennent* » [62] (article L611-19). Whereas this clause constitutes a strong political signal, its scope is limited to patents delivered within the French territory, (these are delivered by the National Institute of Intellectual Property (INPI). See https://www.inpi.fr/fr), which only relocates the problem—French seed business may still appeal to the European Patent Office to obtain intellectual property rights at the European scale.

### 3.3.2. The Limits of the Intellectual Property Regime

Despite the recent reforms on the French side, the lenient legislation on seed breeding is regularly pinpointed as a leading cause of the increasing market concentration of the sector [31]. Moreover, current seed laws are arguably becoming less and less adapted to the challenges of the 21st century; a previous study [63] showed that the intellectual property regime instituted since UPOV 1961 has favoured genetic engineering, while locking out agroecological innovations. Yet, this might become problematic when taking into account that the productivity gains of biogenetics and other artificial inputs have been nil for over two decades. Furthermore, the risks posed by the privatization of native genes challenges the soundness of the intellectual property system as such.

Another previous study [64] argues that contrary to the argument of the "tragedy of the commons", problems relating to the common use of resources could be resolved more efficiently by voluntary organization than by coercive measures or privatization, provided that transaction costs are sufficiently low, so as not to undermine negotiation processes. Where competition is abolished, the fear of illegitimate appropriation vanishes, and intellectual property rights become devoid of meaning. In fact, the sharing of knowledge and genetic material, such as promoted by the *Réseau Semences Paysannes*, may be a better catalyser of progress than the traditional intellectual property system, as it fosters strong synergy, collective problem-solving, and extensive feedback loops. For instance, studies have empirically confirmed that mutual help and the systematic diffusion of knowledge are contributing factors of the superior profitability of permaculture (permaculture is an agroecological practice which aims to mimic as closely as possible patterns that can be observed in nature. It is characterized, among other things, by a specific circular design, the respect of the lunar calendar, and the absence of ploughing), compared to conventional market gardening practices [65,66].

### 3.3.3. The 2004 International Treaty on Plant Genetic Resources for Food and Agriculture, and France's Position

In the political realm, this alternative perspective has historically been endorsed by the United Nations. In the "International Treaty on Plant Genetic Resources for Food and Agriculture" [67], the Food and Agriculture Organization (FAO) enjoins heads of governments to "promote the collection of plant genetic resources", and "support . . . farmers and local communities' efforts to manage and conserve on-farm their plant genetic resources" (article 5(1)). Moreover, the treaty recognizes "the enormous contribution that the local and indigenous communities and farmers of all regions of

the world have made, and will continue to make, for the conservation and development of plant genetic resources".

From this acknowledgement two fundamental rights for the peasant community have been derived. First, "the right to equitably participate in sharing benefits arising from the utilization of plant genetic resources for food and agriculture", and second, "the right to participate in making decisions, at the national level, on matters related to the conservation and sustainable use of plant genetic resources for food and agriculture" (article 9.2(b) and (c)). The treaty is deemed to be binding ("Each Contracting Party shall ensure the conformity of its laws, regulations, and procedures with its obligations as provided in this Treaty" (article 4)), but as often the case in international law, it lacks enforcement mechanisms.

France promptly ratified the treaty in the year of its redaction, but more than ten years later, many promises remain unanswered, when not discarded by antagonistic national laws. Admittedly, the 2016 *Loi pour la Reconquête de la Biodiversité, de la Nature et des Paysages* [61] legalized the exchange of peasant seeds, and restricted the scope of the breeders' rights so as to prevent the privatization of native genes, placing France in the vanguard of the subject. However, the country is the only one of the European Union to levy a "solidarity" tax on farm-saved seeds. More recently, Macron's government launched a call for projects, with a budget of 300,000€, to support initiatives promoting "the characterization and sustainability of collections of plant genetic resources for agriculture" ("*Initiatives portant sur la caractérisation et la pérennisation des collections de ressources phytogénétiques de plantes cultivées* », translation from the authors) [68]. Nonetheless, central government support for improving the adaptability and resiliency of cultivated species remains quite limited. The National Institute for Agronomic Research (INRA) counts no more than 15 Centres of Biological Resources (CBRs), unevenly scattered over the territory. Each of these centres is specialized in the collection of one or two species, totalling more than 500 varieties in the aggregate, and 217,826 samples of seeds [69]. However, they are currently only accessible to peasant networks in the context of participatory research [70–73].

## 4. Discussion

### 4.1. Current Policy Developments in France

France's current president (2017–2022), Emmanuel Macron, promotes and supports the genetic diversity of cultivated species. The minister of agriculture, Stéphane Travert, recently claimed that France bestowed a "great importance to the Treaty (on Plant Genetic Resources for Food and Agriculture)" [68], stressing that the adaptation of agriculture to climate change was a major challenge [68]. These words suggest that the current government intends to bring the French legislation in-line with the precepts of the FAO.

Mr. Travert has also shown to be preoccupied by the precarious status of farmers [74]. While it is welcome that the crucial role of peasant seed networks in biodiversity conservation is politically recognized, it is essential to emphasize that scaling up their use may also improve the stability of the farming profession as a whole, while not necessarily compromising food security and the French commercial balance, provided that more extensive research is conducted.

Supporting the development of peasant seed networks, although being a symbolic step, is only the timid beginning of the transition towards agroecology. Solving the social, economic, and climatic challenges French agriculture is facing today necessitates a holistic approach, which goes beyond the scope of this paper. For a more comprehensive account of the paths to reforming the agricultural system, a prior study [75] theorized the agroecological transition in France.

### 4.2. Policy Recommendations

In light of the above discussion, the paper gives five policy recommendations. First, France should keep the commitment to respect peasants and farmers' integrity and sovereignty, as enshrined in article 9.2 (c) of the Treaty on Plant Genetic Resources for Food and Agriculture [67]. This implies

strengthening the dialogue with conventional and alternative cultivators, notably the *Réseau Semences Paysannes*, in order to increase the transparency of agricultural policies. This is in line with previous researchers' [20] argument that farmer's autonomy has to be protected in order to advance organic agriculture in particular.

Secondly, and in the light of the first recommendation, article 16 of the *Loi Relative aux Certificats d'Obtention Végétale* [59], concerning the "obligatory voluntary contribution", an oxymoron in itself. should be revoked. A similar point is made by other researchers [18,19], who argue that reuse of seed varieties is often diminished because of the establishment of seed laws which favor commercial seed. While the right for farmers to reuse their seeds is a slightly different problem than that of peasant seeds, it is a first step towards cultivators' empowerment.

Third, the exchange of peasant seeds should be fostered through the creation of regional and sub-regional seed banks. As previously mentioned, CBRs are today only accessible for research purposes. Moreover, their limited number and uneven distribution within the national territory restricts their role to ex-situ conservation means. By increasing their number, diversification, and improving their accessibility, CBRs could ensure an equitable distribution of free genetic resources within the French territory, thereby improving peasants' access to landraces and preventing the risk of seed shortages. In addition, CBRs could play an important role in reducing peasants' information costs (i.e., scientific and technical difficulties) through logistical assistance (adapted materials, knowledge on the cultivation of landraces, etc.). This is in line with a previous study [76], which provides concrete examples of how seed diversity can directly be associated with improving capability to feed an increasing population, and therefore have a direct effect on health.

Fourth, the divergence in views between Paris and the European Union as regards plant patenting undermine the French efforts to prevent the appropriation of native genes. A harmonization of the European legislation would be ideal. However, this prospect will likely be a delicate exercise, given the plurality of the other Member States' legislative framework, [77].

Lastly, research on the channels through which agricultural techniques affect the genetic expression of plant varieties should be encouraged. Increasing funding in order to multiply participatory research initiatives could help remedy the current uncertainty as regard the factors influencing the yield of landraces, while helping peasants improve their breeding techniques and develop optimal agroecological practices tailored to specific varieties.

## 5. Conclusions

The French agricultural model seems, despite its past benefits, unable to answer the challenges of the 21st century. Over the last two decades, the joint effects of falling productivity gains, volatile food prices, and rising input costs have weighed heavily on the standard of living of cultivators and the profitability of the sector, leading to the suggestion of a "rupture in technical progress". In addition, the current intellectual property regime established following the end of the second world war has prioritized the development of homogenous high yield cultivars, reinforcing the market power of a few seed breeders. As a consequence, crops' genetic bases have gradually narrowed, limiting plants resiliency to climate change and other hazards.

In response to these economic, social, and environmental challenges, a growing number of scientists advocate the adoption of agroecological farming practices. Changing agricultural models, however, implies rethinking the conventional system of seed production. In this regard, our analysis of alternative seed procurement networks ("peasant seed houses") on the French territory suggests that peasant seeds network may effectively contribute to overcoming many blockages in the agricultural world. First, peasant seeds, by virtue of their greater genetic diversity, ensure more stable production and have greater adaptive capacities. Second, peasant seed houses create social links, which may alleviate farmer's psychological burden. Lastly, the free exchange of peasant seeds makes farmers less dependent on inputs, which may prevent dangerous debt cycles.

These benefits have to be balanced against the inherent limitations of the peasant selection system. These include higher information and supervisory costs, as well as a risk of lower yield. Furthermore, the current legislative context represents another hindrance to the development of peasant seed houses, though France has recently taken steps to remedy the situation.

Nonetheless, these limitations could be partially or totally eliminated if adequate policies are implemented. In this regard, the recommendation is to: (i) strengthen the dialogue with farmers in the shaping of policies related to of use of plant genetic resources; (ii) abrogate the "obligatory voluntary contribution" on farm-saved seeds; (iii) diversify the collection of Centres for Biological Resources, increase their number, and democratize their access; (iv) harmonize the French and European Union regime on intellectual property; and (v) encourage participatory research. This latter measure is perhaps the most crucial to the generalization of the use of peasants' seeds. More research needs to be conducted on landrace improvement techniques in support of a consistency among rusticity and ecology and profitability.

This paper fills in a gap in the literature by analyzing the economic consequences of gratuitous seed exchanges, which were legalized in France in 2016. By investigating the promises and challenges of scaling up the use of peasant seeds in France in the context of the agroecological transition, this article contributed to the economic literature in two ways: First, by exploring non-monetized seed exchanges, a perspective which echoes their status of "common heritage of humankind", such as defined by the Food and Agriculture Organization [16] (p. 165); and second, by concentrating the analysis at the national level, for a country which has a strategic position on the global seed market.

Our paper contributes to the literature on peasant seeds by providing an analysis of the current position of France. By analyzing data on recent time-series data (related to agricultural productivity, prices, costs, revenues, and indebtedness) with a description of the legal perspective and an interdisciplinary literature review, we are able to trace out objective policy recommendations for France. Suggestions for future research include expanding the analysis to other countries, to European Union countries in particular, and the impact that trade relations have on the erosion of cultivated species and biodiversity. Research is also needed to understand the development of participatory plant breeding projects, while considering the obstacles they face and achievements they have already made. From a legal perspective, there is the need to map the differences across countries with respect to the laws in place to promote cultivated species diversification and to promote a model for seed procurement. Some initiatives are already in place, as this paper showed in Section 1, but a more in-depth analysis would contribute to the advancement of a more resilient agriculture at a larger scale.

**Author Contributions:** C.G., H.F.M.W.v.R., and J.S. designed the research. C.G. (she was the first author) set up the conceptualization and analyzed the literature. H.F.M.W.v.R. and J.S. reflected upon the results. C.G. prepared the original draft. H.F.M.W.v.R. and J.S. reviewed and edited the final version of the paper. All authors contributed to the writing of the paper. All authors read and approved the final manuscript.

**Funding:** This research received no external funding.

**Conflicts of Interest:** The authors declare no conflict of interest.

## Appendix A

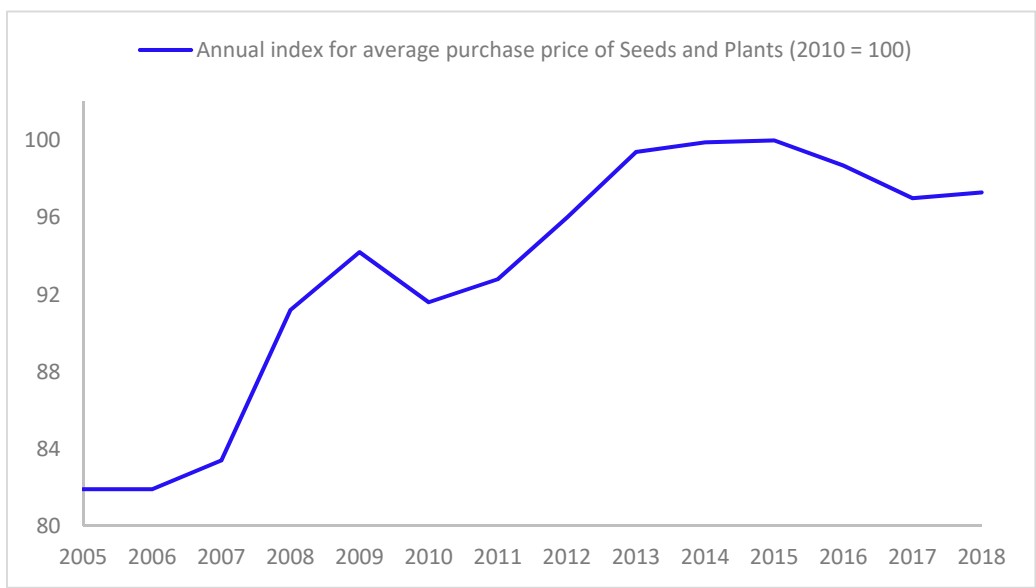

**Figure A1.** Agricultural means of production purchasing price index (IPAMPA), Seeds Cost Index. Notes: Computation based on the data from the Insee.

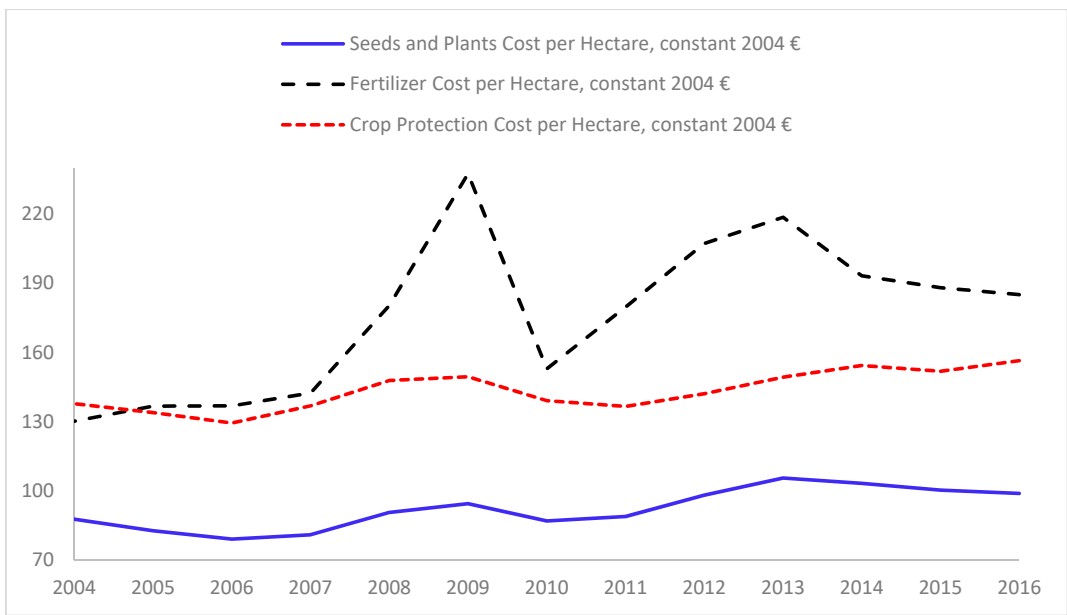

**Figure A2.** Specific crop costs on the rise. Notes: Computation based on the data from the FADN.

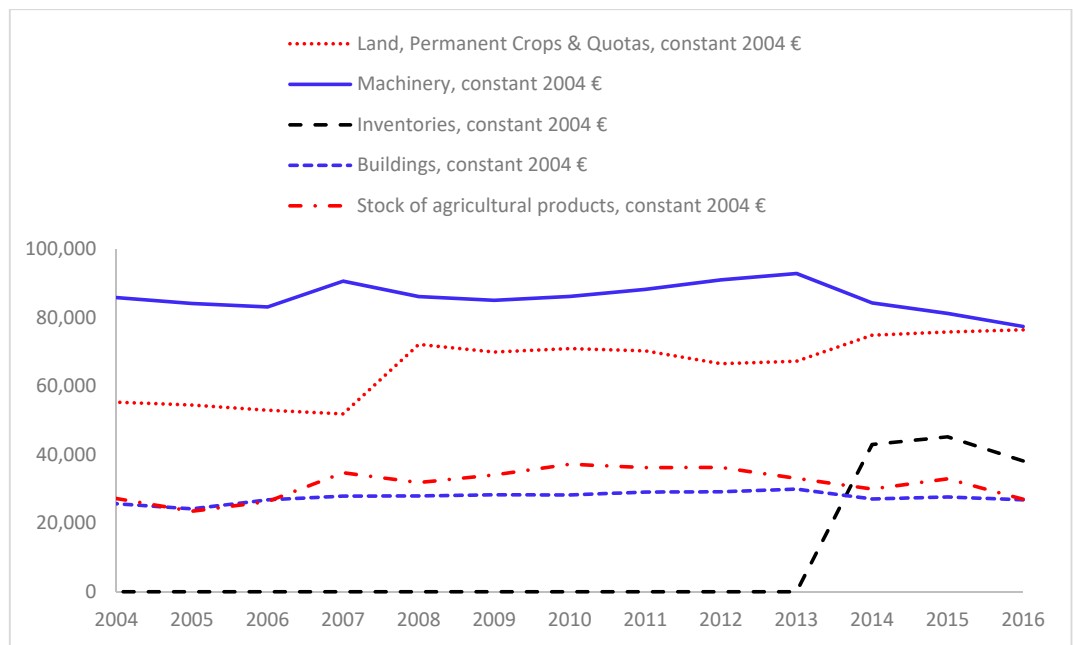

**Figure A3.** Land acquisition and build-up of inventories. Notes: Computation based on the data from the FADN.

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
