# Peer review of "Peasant Seeds in France: Fostering A More Resilient Agriculture"

_sustainability, doi:10.3390/su11113014_

Round 1
Reviewer 1 Report
The paper has improved significantly and all of my previous comments are addressed, however, I have a few minor comments as follows:
- I believe merging the Introduction section and the Methods section is not a good idea, please provide a separate Methods section and move the relevant inputs to this section.
- Adding a few sub-sections might improve the overall structure of the Discussion section.
- Also, the author should separate the different topics into the same paragraphs. Currently, the same paragraph deals with various topics the Discussion section.
- The authors should address future research directions in the Conclusion section in one paragraph.
Author Response
We followed all suggestions from reviewer 1 and have thus:
- Disaggregated the introduction section into two parts: introduction and methods section;
- Added two sub-sections to the Discussion section. We start this section by presenting a few political positions in France with respect to the support of cultivated species diversity and follow by presenting five policy recommendations. Thus, now we created a sub-section entitled Current policy developments in France and a second sub-section entitled Policy recommendations
- We separated paragraph 2 in the Discussion section into two. The first one focuses on the status of the farmers; and the second one focuses on the challenges of transitioning towards agro-ecology.
- We have now added a paragraph at the end of the Conclusion section with a few directions for future research.
Reviewer 2 Report
The work is interesting, focusing on the positive and negative aspects associated to the development of alternative seed procurement networks in France; but it needs to be reviewed from the point of view of the structure.
A part of the methodology used is missing. For this reason, it is necessary to insert it before moving on to the results. Moreover, it is important to better describe the context of analysis (France) in order to frame this theme well by describing the past and the current state in a single paragraph.
While the part of the results can be accepted, the part of the discussion needs to be increased. What implications does your work have in the scientific world? What originality do you highlight? This can also be linked to the methodology used as it is now difficult to make a comment as it is missing.
Author Response
We followed all suggestions from reviewer 2 and have thus:
- Made a separate section entitled Methods
- Added a paragraph on France to contextualize the past and current state. This paragraph was added at the end of the sub-section “The French Agricultural Landscape”.
- We added the contribution of our paper to the conclusion section, by linking it to possible future research lines.
Round 2
Reviewer 2 Report
The article has improved. I suggest to revise the English language and add a conclusion part with a synthetic final comment.
Author Response
We have revised the text and added a paragraph placing our paper into a broader literature persepctive.

This manuscript is a resubmission of an earlier submission. The following is a list of the peer review reports and author responses from that submission.
Round 1
Reviewer 1 Report
Comments:
1. The manuscript addresses an interesting subject.
2. In line 15 of the abstract, it is mentioned that “We investigate the positive and negative aspects…”. The authors should avoid the use of personal pronouns within the body of the paper (e.g. "this paper investigates..." is correct; "I/we investigate..." is incorrect). Please revise the whole paper.
3. The authors should change “agriculture” to “resilient agriculture” and add “biodiversity” as the last keyword.
4. The authors have defined the term “agroecology” in a wrong way; it is not about organic agriculture. Based on the FAO definition “Agroecology is a scientific discipline, a set of practices and a social movement. As a science, it studies how different components of the agroecosystem interact. As a set of practices, it seeks sustainable farming systems that optimize and stabilize yields”. The authors should revise the footnote according to this definition in page 1.
5. The authors have added unusual numbers of footnotes in page 2. I suggest adding them to the main text. Please do it for the whole manuscript.
6. To enrich the Introduction section the authors should mention a few relevant/recent studies along with their approaches and outcomes and highlight the main contribution of the current study by comparing it with previous ones.
7. The authors should explain whether it was a specific reason for choosing the duration of 2004-2016 as the time period in the Methods section.
8. The Methods section is very short; the authors should enrich this section by elaborating more on their approaches.
9. The subsection 3.1.5 is very short; the authors should either add more explanation or merge the inputs with other subsection in the Result section.
10. To improve the Discussion section the authors should outline how the main findings are in line with previous studies. In other words, the authors should add relevant literature reviews which gained the similar results and discuss how their findings are consistent with the current study.
11. The authors should highlight the main policy implications of the current study in the Conclusion section. It should be done in 1-2 paragraphs.
12. The authors should address future research directions too. It could be done in one paragraph in the Conclusion section.
Reviewer 2 Report
The paper presents relevant methodological mistaques and a considerable weak review.
General empirical conclusions highlight the problems of the presented paper.
Reviewer 3 Report
The work is interesting and current, however it lacks a scientific structure.
The introduction is the only part built ad hoc, with good information with comparison also with the United States.
For the rest, it seems all a review as a collection of data until today. A real interdisciplinary method seems to be missing. The results are taken from other work or simple elaborations without putting them together and making a real, scientific contribution.
I advice to totally rewrite from the part of the methods onwards, actually highlighting your real findings